# Non-parametric Uncertainty Assessment in Deep-Learning Based Affine Image Registration

No Author Given

No Institute Given

**Abstract.** Affine image registration plays a key role in diagnosis, surgical planning and in data-processing pipelines for research as both an essential initialization for subsequent non-rigid registration or as an independent step. Uncertainty quantification in deep learning (DL) based image registration models is critical for determining confidence intervals required for surgical guidance, and for reliable assessment of differences between the registered images. We introduce AIR-SGLD - a non-parametric fully Bayesian framework for Affine image registration. We use Stochastic gradient Langevin dynamics (SGLD) during the training phase to characterize the posterior distribution of the network weights. We demonstrated the added-value of AIR-SGLD on the brain MRI (MGH10) dataset in comparison to the baseline AIR DL-based Affine image registration framework using 300 pairs of images generated from the MGH10 dataset. Our experiments show that AIR-SGLD outperforms AIR by means of cross-correlation between the images (0.91 vs. 0.87, $p < 0.01$). Further, AIR-SGLD provides an estimate of the registration uncertainty that correlates with both registration error (Pearson correlation coefficient of $R = 0.769$) and the presence of out-of-distribution data ($R = 0.796$). AIR-SGLD has the potential to provide reliable and more accurate registration for clinical diagnosis, surgical planning, and automatic data processing pipelines.

**Keywords:** Affine Registration · Brain MRI · Uncertainty Assessment

## 1 Introduction

Registration is the process of mapping a pair of images (e.g MR images acquired from different subjects) onto one coordinate system [18]. It is a fundamental task needed in a wide-range of medical and neuro-science applications including diagnosis, response to therapy assessment, surgical planning and in automatic imaging data-processing pipelines.

Specifically, Affine registration plays a key role either as the main task or as an essential initialization for deformable image registration [14, 11].

Classical methods tackle the registration task by formulation of an optimization problem over the transformation parameters and solve it iteratively [12, 7, 13]. In these approaches, a similarity metric, which is measured between the two images to be registered, is optimized. However, these conventional algorithms are computationally demanding, which in turn, makes the registration of a new pair of images a computationally expensive process.

In light of the success of DL-based methods in numerous computer vision tasks, several studies aimed to propose more efficient and less time-consuming registration approaches based on DL models [5, 2, 17, 15, 3]. These techniques predict the registration model either through a supervised learning framework (i.e with the help of provided reference deformation fields) [17] or in an unsupervised manner [2, 1, 15].

However, such DL-based image registration methods lack computational mechanisms for quantifying the risks of failure in their predictions. Quantification of the registration error and providing uncertainty measures about the transformation model has significant practical utilization in medical and research applications for which confidence intervals assessment is required.

Bayesian DL-based models have the potential to enable safer utilization in medical imaging, improve generalization, and assess the uncertainty of the predictions by characterizing the entire posterior distribution of the network parameters. Recently, two main methods were proposed to assess the uncertainty in DL-based methods for medical image registration [1, 17]. The approaches include inference-time dropout [8] and variational encoder-decoder models [9]. These approaches, however, are limited to specific DL architectures. In addition, the latter assumes a parametric distribution of the latent space in the form of a Gaussian distribution which may represent an oversimplification of the unknown true underlying distribution.

In this paper, we develop AIR-SGLD - a new, non-parametric, Bayesian DL-based approach for Affine registration of brain MRI images. The proposed DL-based system predicts the posterior distribution of the transformation parameters. We achieve this by adopting the strategy of Stochastic gradient Langevin dynamics (SGLD) [16] to sample from the posterior distribution of the network weights [4]. The posterior distribution of the transformation parameters of the Affine matrix is then used to provide an uncertainty assessment of the prediction.

Our registration system's backbone is based on the architecture of the AIR-Net Affine registration model [3]. Specifically, we inject Gaussian noise to the loss gradients during the training phase of our framework, and keep all weights obtained after the *"burn-in"* iteration in which the training loss curve exhibits only small variations around its steady-state. At inference time, we estimate the statistics of the predicted 3D Affine transformation by averaging predictions obtained by the model with the saved weights.

Our experiments were performed on 300 image pairs generated from brain MRI images belonging to the MGH10 database [10]. We demonstrate that AIR-SGLD outperforms the baseline AIRNet by means of registration error (means-square-error MSE of 0.0016 vs. 0.002, $p < 0.01$). Further, AIR-SGLD measurements of uncertainty correlated with both registration error (Pearson correlation coefficient of $R = 0.769$) and presence of out-of-distribution data ($R = 0.796$).

Our non-parametric Bayesian approach implemented in AIR-SGLD does not make any prior assumptions about the underlying distribution and can be applied to any DNN architecture. Further, it can be adjoined to most DNN training

schemes, either supervised or unsupervised. Thus, it has the potential to enable a safer utilization of DNN-based methods in safety-critical applications.

## 2    Background: Deep-learning based Affine Registration

In this work, we address Affine image registration. The main task is to predict a 3D Affine matrix to parameterize the transformation that maps the pair of input images.

Let $I_M$ and $I_F$ be the 3D moving and fixed input images, respectively. $a$ is the twelve-dimensional vector to flatten rows of the Affine matrix $A$. The Affine registration task can be formulated as a prediction by a trained model:

$$a = f_\theta(I_M, I_F) \tag{1}$$

where $\theta$ are the parameters of the network that are obtained after training the model, by optimizing the following:

$$\hat{\theta} = \arg\min_\theta L(I_F, I_M \circ \Phi_A) \tag{2}$$

where $I_M \circ \Phi_A$ denotes the result of warping the moving image with the Affine transformation parameterized by $A$, $\Phi_A$. $L$ is a dissimilarity term, which quantifies the resemblance between the warped and the fixed input images. The aforementioned kind of optimization provides the best point-estimate of network parameters, which is the solution of Maximum Likelihood Estimation (MLE) rather than characterizing the entire posterior distribution of the network parameters.

## 3    Proposed Approach

### 3.1    Non-parametric Bayesian Affine image registration

Our goal is to characterize the posterior distribution of the Affine transformation parameters:

$$\hat{\theta} \sim P\left(\theta | I_F, I_M\right) \propto P(I_F, I_M | \theta) P(\theta) \tag{3}$$

Since direct integration of the posterior distribution is intractable, we efficiently sample the actual posterior distribution of the model weights using an *adaptive* SGLD mechanism. We treat the network weights as random variables and aim to sample the posterior distribution of the model prediction. To this end, we incorporate a noise scheduler that injects a time-dependent Gaussian noise to the gradients of the loss during the optimization process. At every training iteration, we add Gaussian noise to the loss gradients. Then, the weights are updated in the next iteration according to the "noisy" gradients. This noise schedule can be performed with any stochastic optimization algorithm during the training procedure. In this work we focused on the formulation of the method for the `Adam` optimizer.

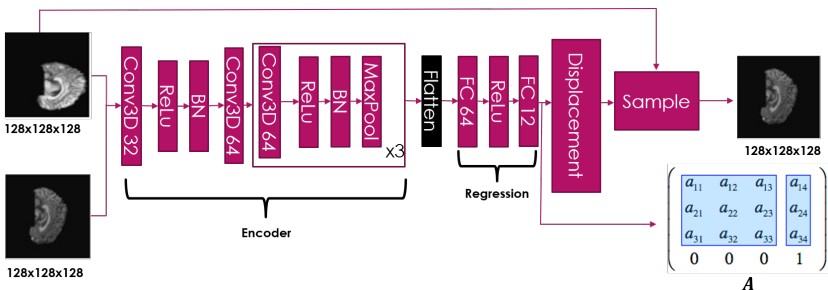

Fig. 1: Block diagram of the proposed registration system. Moving and fixed images, $I_M$ and $I_F$, are concatenated to be a two-channel input that is fed to the *encoder* block. Spatial features encoded by the *encoder* are then flattened and used as an input to the *regression* unit. The regression block predicts the parameters of the Affine matrix. Lastly, the moving image is sampled with the corresponding Affine transformation, which is parameterized by the predicted matrix, to yield the resulting registered image as an output.

Let $L(I_M, I_F, f_\theta(I_M, I_F))$ denote the overall registration loss, described in (2). We denote the loss gradients by:

$$g^t \overset{\triangle}{=} \nabla_\theta L^t(I_M, I_F, f_\theta(I_M, I_F)) \tag{4}$$

where $t$ is the training iteration (epoch). At each training iteration, a Gaussian noise is added to $g$:

$$\tilde{g}^t \leftarrow g^t + \mathbf{N} \tag{5}$$

where $\mathbf{N}^t \sim \mathcal{N}(0, s)$, $s$ is a user-selected parameter that controls the noise variance (can be time-decaying or a constant).

Lastly, we save the weights of the network that were obtained in iterations $t \in [t_b, N]$, where $t_b$ is a pre-determined parameter of the SGLD-based method and $N$ is the overall number of iterations. It is essential to select a $t_b$ larger than the cut-off point of the *burn-in* phase. One should sample weights obtained in the last $t_b, .., N$ iterations, where the loss curve has converged.

### 3.2   Network Architecture

Fig. 1 describes the Architecture of our system. Our main building-block is a convolutional network (CNN) similar to the AIRNet model [3]. The pair of fixed and moving images, $I_F$ and $I_M$, respectively, are concatenated to be a two-channel input. The model is composed from a CNN-based *encoder* and *regression* block. The *encoder* extracts spatial features from each input image, $I_F$ and $I_M$, which are flattened and then used as an input to the regression block to estimate the Affine transformation $A$. Then, the spatial Affine transformation, $\Phi_A$, is calculated from $A$ and used to warp $I_M$.

The *encoder* includes 4 CNN-based blocks. The first block is comprised from a 3D convolutional layer with feature size of 32 (Conv3D 32) followed by ReLu activation, a Batch normalization (BN) layer and another 3D convolutional layer with feature size of 64 (Conv3D 32). The three later blocks each include a 3D convolutional layer with feature size of 64 (Conv3D 32) followed by ReLu activation, a Batch normalization (BN) layer and max pooling of size 2. Kernel size of all convolutional layers is $3 \times 3 \times 3$ and a stride of 2 is used.

The *regression* block is comprised from two fully connected layers of features sizes 64 and 12, where the first is followed by the ReLu activation function. The last layer has 12 neurons since it predicts the 12 elements of $A$, which parameterize the corresponding Affine transformation.

### 3.3   Inference

Having the network trained, and its weights during training saved, we exploit only the outputs of the network with weights obtained after the *burn-in* phase, i.e. in the last $t_b, .., N$ iterations.
We sample a set of predictions $\{A^t = f_{\theta^t}(I_M, I_F)\}_{t_b}^N$, obtained by feed-forwarding the pairs $I_M, I_F$ to DNN-based registration models with weights that were obtained in the last $t \in [t_b, N]$ iterations. Then, when we have a new pair for alignment, we estimate the averaged posterior Affine matrix

$$\overline{A} = \frac{\sum_{t=t_b}^N A_t}{N - t_b} \tag{6}$$

In addition, we quantify the variance of the Affine matrix, which is used to characterize the uncertainty of registration. Lastly, we register the moving image by resampling its coordinate system with the Affine spatial transform $\Phi_{\overline{A}}$, characterized by the averaged matrix $\overline{A}$.

## 4   Experiments and Results

*Dataset* In our experiments we used the MGH10 brain MRI [10] database. The MGH10 dataset consists of brain MRI scans of 10 subjects with provided segmentations into 74 regions. MRI images were Affine-registered according to the MNI152 template [6], and preprocessed by inhomogeneity-correction. All scans are of size $182 \times 218 \times 182$ and uniform spacing grid of $1mm$ in each dimension. All images in the dataset were resampled to a uniform size of $128 \times 128 \times 128$. We generated 300 pairs of fixed and moving images by resampling each moving image in the dataset with 30 random Affine matrices, which were generated to construct the corresponding 300 fixed images. Then, we randomly split the pairs of moving and fixed images to training and test sets, which includes 270 and 30 pairs, respectively.

### 4.1   Evaluation methodology

To conduct a quantitative comparison, we trained two models, which have the same architecture described in section 3.1 and used Normalized Cross Correlation (NCC) as the training loss. Both models were trained with the same settings, but the first is based on SGLD, i.e. with incorporation of noise to the gradient, whereas the latter is trained without adding noise. In our experiments, we selected a fixed standard deviation, $s = 0.001$ for the injected noise variance. The network parameters are then updated according to the `Adam` update rule. We used the `Adam` optimizer with learning rate of 0.0001 for both models. Hence and henceforth, we refer to the first model as AIR-SGLD and the second as AIR-Adam. In both models, the network weights were initialized randomly, except for the regression layers, which were initialized by zero weights and identity transform bias.

For each pair of images in the test set we sampled predictions of the AIR-SGLD network with weights obtained at the last 10 iterations. We then calculated the average Affine matrix, which was used to resample the moving image to yield the resulting registered image. In addition, we estimated the variance of the Affine matrix from the 10 samples, which was used to assess the uncertainty of the registration.

We assessed the accuracy of the registration models by calculating the mean square error (MSE) between the estimated Affine matrix and the ground truth, which is predetermined and used to generate the fixed image, for our AIR-SGLD method and the benchmark AIR-Adam. Further, to quantify the similarity between the resulting registered image and the fixed one, we calculated the cross correlation (CC) metric between the registered image and the fixed image for all pairs of images in the test set.

We evaluated the correlation between the uncertainty measures produced by AIR-SGLD and 1) the presence of out-of-distribution data, and 2) registration error as follows. We generated out-of-distribution images by adding Gaussian noise with increasing standard deviations, $\sigma_n$, to the pair of input images. We calculated the variance of each parameter in the transformation and averaged over the transformation parameters to get a scalar value representing the overall uncertainty in the transformation estimation (denoted later by $\widehat{\mathrm{Var}}(A)$).

### 4.2   Results

Fig. 2 presents examples of registration results of both our proposed AIR-SGLD and AIR-Adam models.

*Registration accuracy* Table 1 presents the mean MSE and NCC, calculated over the whole test set, for both models. We observed, AIR-SGLD achieved significant improvements over AIR-Adam in terms of MSE and CC ($0.00165, 7e^{-4}$ vs. $0.0022, 0.0016$, with paired t-tests of $p = 0.0082$, $p = 0.00056$ for MSE and 0.917 vs. 0.878 with p-value of $1.820e^{-7}$ for CC).

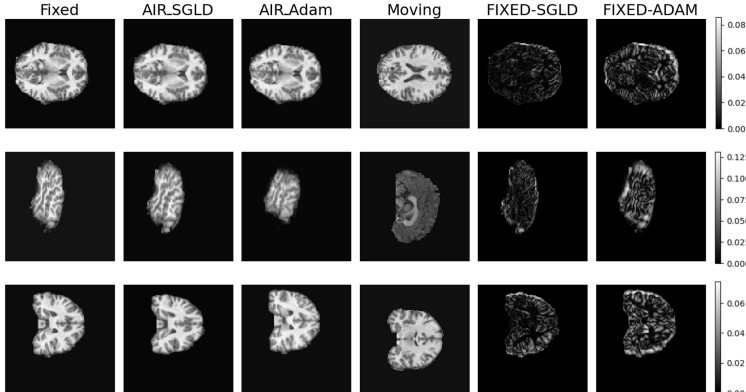

Fig. 2: Registration results. from left to right: $I_F$, the resulting registered image, obtained by our method AIR-SGLD,$I_{SGLD}$, by AIR-Adam,$I_{Adam}$, $I_M$ and the Affine matrix used in resampling $I_M$ to generate $I_F$. Cross correlation (CC) calculated between $I_f$ and the registered images obtained AIR-SGLD and AIR-Adam, $I_{SGLD}$ and $I_{Adam}$, for the three examples are: 0.89 vs. 0.86, 0.926 vs. 0.920 and 0.924 vs. 0.846, respectively.

Table 1: Registration Accuracy. Mean and std values of MSE and CC for both AIR-SGLD and AIR-Adam methods. Mean values of MSE, between the predicted and ground truth matrix, are calculated for the entries of the linear matrix (first three columns) and the translation vector (last column) independently.

|  | MSE | | CC |
|---|---|---|---|
|  | **Linear** | **Translation** |  |
| **AIR-SGLD** | $0.00165 \pm 0.0011$ | $7e^{-4} \pm 8e^{-4}$ | $0.9178 \pm 0.031$ |
| **AIR-Adam** | $0.0022 \pm 0.0015$ | $0.0016 \pm 0.0015$ | $0.8783 \pm 0.041$ |

As one can infer, the CC of AIR-SGLD is higher, which indicates the resulting warped image obtained by our method is more similar than that of AIR-Adam to the fixed image.

*Uncertainty Assessment* The correlation between $\widehat{\mathrm{Var}}(A)$ and the level of data corruption by noise and registration error are depicted in Fig. 3. Pearson correlation coefficient calculated between the measure of uncertainty and each one of the MSE and noise levels are: $R = 0.769$ and $R = 0.796$. This, in turn, indicates the ability of the uncertainty measure to detect out-of-distribution data that would result in unreliable registration performance.

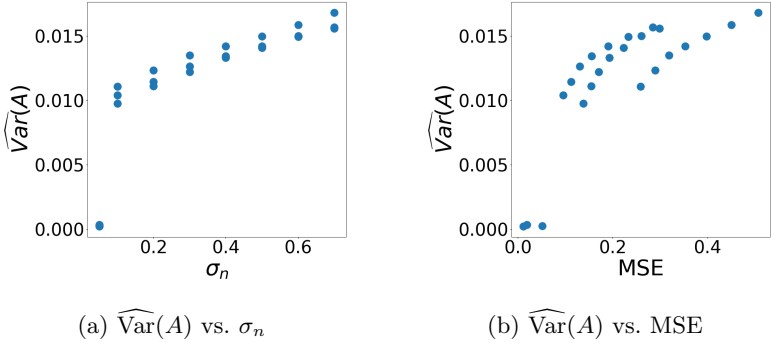

(a) $\widehat{\mathrm{Var}}(A)$ vs. $\sigma_n$                    (b) $\widehat{\mathrm{Var}}(A)$ vs. MSE

Fig. 3: Uncertainty Assessment. (a) and (b) Scatter plots of the mean value of variance estimate of $A$, $\widehat{\mathrm{Var}}(A)$, versus the amount of noise added to the input images and the MSE, calculated between $\bar{A}$ and the ground truth $A$, respectively.

## 5    Conclusion

In this paper we developed AIR-SGLD, a non-parametric Bayesian DNN-based method for Affine registration of brain MRI images. Specifically, we used noise injection for the training loss gradients to efficiently sample the true posterior distribution of the network weights. In this work, our training relies on an `Adam` optimizer, however, it can be directly extended to other optimization techniques as well. In this work, AIR network was used as a baseline to illustrate the effectiveness of training with SGLD and its ability to produce a clinically significant uncertainty measure. However, the proposed technique is not limited to a specific architecture and can be incorporated to any existing network. The conducted experiments showed that AIR-SGLD improves the registration accuracy, measured by the MSE between the predicted and the ground truth Affine matrix, compared to training the system without noise incorporation. Further, it enables uncertainty quantification, where the measured uncertainty of AIR-SGLD correlates with registration error and out-of-distribution data.

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
