# OpenReview forum: "Non-parametric Uncertainty Assessment in Deep-Learning Based Affine Image Registration"
_WBIR.info/2022/Workshop/Biomedical_Imaging_Registration — Reject_

### Official Review · Reviewer_EcEy · 2022-02-12

**Rating:** 2
**Confidence:** 5

**Deanonymize Review:**

no

**Detailed Comments:**

Comment:

A more common application of affine registration is the atlas-based registration, which registers the target scan to a well-delineated/normalized template. Extending the experiments to atlas-based registration or inter-subject registration will be beneficial to this work.

Open-source, large-scale brain MRI datasets such as OASIS, Mindboggle-101 can be used for the experiment.

Apart from the MSE and CC of the registration results, one can measure the overlapping anatomical structures of the registered brain scans to quantify the registration accuracy, which helps to improve the clinical relevance of this work.

This paper will be beneficial to include a paragraph explaining how the estimated uncertainty of the registration can be used in image registration or its downstream analysis.

---------------------
Overall, this paper proposed an interesting method for joint affine registration and uncertainty estimation. However, the experiment setting is problematic, and the method is not adequately evaluated.


**Paper Type:**

methodological development

**Strengths Weaknesses:**

Summary:
This paper proposes an unsupervised learning-based affine registration method with a non-parametric Bayesian framework. The affine registration is parametrized with a CNN and the network weights are regarded as random variables, which aims to sample the posterior distribution of the model prediction. During training, a gaussian noise is added to the gradient of the loss. The final prediction of the affine matrix is the average predictions of CNN models with weights saved at various training epochs. The uncertainty of the registration is quantified by the variance of the resulting affine matrix. The method is evaluated on affine augmented MGH10 dataset, which contains brain MRI scans of 10 subjects and 300 affine augmented image pairs. The results are compared with a baseline method, which does not add noise to the gradient and show improvement over the baseline method in terms of MSE and cross-correlation (CC) of the registration.

-----------------------
Strengths:

The proposed method is capable of joint estimating the affine transformation and uncertainty estimation, which could be useful in clinical applications.

The problem is well-motivated, and the paper makes a contribution to the learning-based affine image registration literature.

Good reproducibility. The authors provided sufficient details about the models, hyperparameter choices and the public dataset used in the experiment.

The paper is well-written and easy to follow.

----------------------------
Weaknesses:

Insufficient and probably biased evaluation. The proposed method was only evaluated with 10 brain MRI scans of 10 subjects, each with 30 random affine augmentations. More importantly, as mentioned in section 4, the augmented data are randomly split into training and test sets, which may imply that all 10 brain MRI scans are involved in both training and test sets. This experiment results with this setting cannot truly represent the models’ performance to unseen data and the evaluation may cause biased results. Also, testing the model with simulated intra-subject registration (same image scan with affine augmentation) only is not sufficient.

The details of the affine augmentation in the experiment are missing. How are the 30 random affine matrices generated? What are the maximum and minimum degrees of rotation, translation, scaling and shearing?

The experiment result of uncertainty estimation in Figure 3 is unclear. How many image pairs are involved in this experiment? I only see three dots in each noise level in figure 3a). If it is 3, I suggest repeating the experiment for all the testing scans at each noise level.

What are “FIXED-SGLD” and “FIXED-ADAM” in figure 2? They are not the affine matrix as described in the caption.

---

### Official Review · Reviewer_itxk · 2022-02-16

**Rating:** 5
**Confidence:** 4
**Recommendation:** Long Oral

**Deanonymize Review:**

no

**Detailed Comments:**

The paper presents a non-parametric fully Bayesian framework for affine image registration that uses noise injection for the training loss gradients to characterize the posterior distribution of the network weights.
The authors demonstrate that this novel approach improves registration accuracy and enables uncertainty quantification.

Though the authors used the AIR network with Adam optimization as a baseline in this work, the technique is not limited to this constellation. It can be applied to any DNN architecture and extended to other optimization techniques, which will be interesting and valuable to a broad audience.

I really liked the paper, but I missed the motivation and technical details for the core aspect of the presented method.

-	p. 4: Why is Gaussian noise added to the gradients? Please provide motivation why this is a good idea and/or literature references backing up this approach.
-	It is unclear throughout the paper whether $I_F$ and $I_M$ are one image pair or a specific set of images. Please provide details in the paper.
-	p. 6: The authors set the user-selected parameter $s=0.001$. Were other values tested? And why was a fixed parameter chosen rather than a time-decaying? Please comment in the paper.
-	p. 6: The parameter $t_b$ needs to be larger than the cut-off point of the burn-in phase and determines the number of iterations $t$ for which the weights are obtained. It was set to $t=10$ in the experiments. Is there a lower bound for $t$? Please comment in the paper.
-	In source [17] the AIR method was evaluated and compared to other state of the art methods on the MGH10 dataset with regard to mean target overlap. It would be interesting to see if/how much AIR-SGLD can improve those results.


**Paper Type:**

methodological development

**Strengths Weaknesses:**

Strength:
-	clear structure
-	language
-	good overview of previous work and how this work differs
-	step by step explanation of the method
-	figures and tables support comprehensibility
-	the shown training method can directly be extended to other optimization techniques
-	approach is not limited to specific DL architectures and can be applied to any DNN architecture

Weaknesses:
-	some technical details and motivation are left open
-	the choice of parameters for the experiments is given but not explained
-	the ratio between training and test data is unusual (9:1)
-	the presented method is compared with the method it is based on (and the extension of which it represents), but not with other state of the art methods that have previously outperformed the underlying method on the same data set

---

### Official Review · Reviewer_QA1M · 2022-02-16

**Rating:** 1
**Confidence:** 4

**Deanonymize Review:**

no

**Detailed Comments:**

strengths:
- the paper is reasonably well written
- evaluation seems sound


weaknesses:
- there is in my opinion a bit too much motivation about how useful this could be in practice.
- the biggest weakness is a complete lack of context. SGLD had previously been used for rigid registration both within an iterative scheme (https://www.semanticscholar.org/paper/A-Markov-Chain-Monte-Carlo-based-rigid-image-method-Karabulut-Erdil) and a deep learning model (https://arxiv.org/abs/2008.03949). Not only does this paper not discuss differences to previous work, but it doesn't even mention it. The method seems a bit to reinvent the wheel by implementing a well-known method within a popular image registration framework
-  the way the authors measure uncertainty might be meaningless. They compute a variance of a distribution that may not even have a variance and they measure the accuracy by calculating the MSE of the ground truth and predicted affine matrices, which doesn't mean anything

**Paper Type:**

methodological development

**Strengths Weaknesses:**

The paper discusses a Bayesian framework for Affine image registration. Stochastic gradient Langevin dynamics are used during the training phase to characterize the posterior distribution of the network weights. Evaluation is done on the MGH10 dataset.

strengths:
- reasonably well written

weaknesses:
- text could be streamlined
- lack of context and missing discussion of previous work
- chosen uncertainty measurement approach might be meaningless

---

### Official Review · Reviewer_PRvX · 2022-02-20

**Rating:** 4
**Confidence:** 3
**Recommendation:** Short Oral

**Deanonymize Review:**

no

**Detailed Comments:**

Minor:
- Novelty of the paper should be outlined explicitly.
- Motivation for s=0.001 should be described.
- Selection of t_b should be described.
- The choice of the parameters of the affine matrix used for generation of the dataset should be described.
- encoder description: “Conv3D 64“ and “Conv3D 32“ mixed up

Overall, this paper is well-written and the presented method is interesting as it is not limited to a specific network architecture. However, further experiments/evaluation should be performed. Therefore, I would give this paper a 'weak accept'.

**Paper Type:**

both

**Strengths Weaknesses:**

**Strengths:**
- The paper is well-written and well-structured.
- The proposed method is universally applicable for deep learning architectures

**Weaknesses:**
- The evaluation of the proposed method should be more extensive. For example it would be interesting
   * how AIR-Adam performs when calculating the averaged affine matrix (like in AIR-SGLD)
   * how the method performs when using the segmentations that are provided for the dataset for evaluation (e.g. Dice scores)
   * how the method performs on a dataset without synthetically generated fixed images
- Since the authors state that the proposed technique could be incorporated to any network, additional evaluation with other network architectures would be desirable. For example, in  Khawaled et al.’s work (only found on arXiv)
   * Khawaled, Samah, and Moti Freiman. "Unsupervised deep-learning based deformable image registration: a Bayesian framework." arXiv preprint arXiv:2008.03949 (2020)
   * Khawaled, Samah, and Moti Freiman. "NPBDREG: A Non-parametric Bayesian Deep-Learning Based Approach for Diffeomorphic Brain MRI Registration." arXiv preprint arXiv:2108.06771 (2021).

   very similar experiments have been performed with integration of SGLD into VoxelMorph

---

### Decision · Program_Chairs · 2022-02-22

**Decision:**

Reject

**Comment:**

While the reviewers found certain merits and interest in the method, a more thorough evaluation and discussion with respect to related work would have been welcomed. When making our final decisions, the paper decision committee took into account all the review comments and the article itself. We trust that the reviews are helpful for improving the work. The following concerns were raised that prevented us from accepting the article in its current form:
- limited evaluation that only measures MSE of affine transformation parameters instead of TRE or Dice overlap
- missing discussion and/or citation of related prior work on SGLD in DL image registration
- doubts about the meaningfulness of the chosen uncertainty quantification